# Federated Hierarchical Anti-Forgetting Framework for Class-Incremental Learning with Large Pre-Trained Models

## Abstract

Large pre-trained models, such as BERT, have demonstrated strong performance across various tasks. However, they are vulnerable to catastrophic forgetting in incremental learning, particularly in federated settings with non-IID data. Existing approaches, such as knowledge distillation and exemplar replay, partially address this issue but often incur high memory overhead, increase privacy risks, and introduce additional computational burden. To overcome these challenges, we propose FEDHAF, a modular framework for federated class-incremental learning with large pre-trained models. FEDHAF consists of three key components: a frozen feature extractor, a feature adjustment module, and a task-specific head. This structure enables efficient adaptation to new tasks while preserving knowledge from previous ones. We further introduce a two-stage training strategy that separates classifier learning from feature alignment. This strategy combines feature-level distillation with balance regularization, improving knowledge retention without requiring extensive parameter tuning or compromising privacy. Extensive experiments on benchmark datasets, including CIFAR-100, TinyImageNet, ImageNet, and Shakespeare, demonstrate that FEDHAF consistently outperforms state-of-the-art methods.

## 1 Introduction

Large pre-trained models such as BERT (Devlin et al., 2018), GPT (Brown et al., 2020), and ResNet (He et al., 2016) have significantly improved the performance of deep learning in natural language processing and computer vision. These models are trained on large-scale datasets, capturing both semantic and visual features. However, their success depends on access to centralized data, which often contains sensitive or private information. In domains like healthcare and finance, this raises privacy concerns and increases the risk of data leakage. Federated learning (FL) offers a practical solution by allowing multiple clients to collaboratively train models without sharing raw data (McMahan et al., 2017; Nguyen et al., 2021; Tran et al., 2024a; He & Wang, 2024). This decentralized approach enhances data privacy and supports training on distributed and diverse datasets.

Applying FL to large pre-trained models introduces several challenges. In many real-world settings, data arrives continuously, and models must be updated incrementally. Federated class-incremental learning (FCL) addresses this need by allowing clients to receive new classes over time and update their models locally while preserving data privacy. However, the combination of evolving tasks, data heterogeneity, and privacy constraints makes it difficult to retain previously learned knowledge across clients. A key challenge in this setting is catastrophic forgetting, where models lose performance on earlier tasks after learning new ones (He & Wang, 2024; Zhang et al., 2024). In centralized scenarios, this problem is often mitigated by assuming independent and identically distributed (IID) data. In contrast, FL usually involves non-IID data, where each client has its own distinct distribution (Zhang et al., 2024). Clients tend to learn local patterns, and when updates are aggregated, the global model often fails to generalize across all tasks. As a result, catastrophic forgetting becomes more severe (Tran et al., 2024a; Kim et al., 2024).

Existing FL methods, such as FedAvg (McMahan et al., 2017) and FedProx (Li et al., 2020), focus on efficient model aggregation but do not directly address forgetting. In centralized learning, ap-

proaches like knowledge distillation, generative replay, and exemplar replay are commonly used to reduce forgetting. However, these techniques are difficult to apply in FL due to privacy concerns and resource limitations. For example, generative and exemplar replay require access to previous data or synthetic reconstructions, which increase memory and computational costs (Shin et al., 2017; Rolnick et al., 2019b). These limitations highlight the need for new approaches that combine the benefits of large pre-trained models with the constraints of FL to reduce forgetting effectively. Another challenge is the high cost of fine-tuning large pre-trained models, which often have millions of parameters. In FL settings with limited computation, communication bandwidth, and storage, directly fine-tuning these models is impractical. The presence of non-IID data and privacy constraints further complicates this process.

To address these challenges, we propose FEDHAF (**Fed**erated **H**ierarchical **A**nti-**F**orgetting), a framework for federated class-incremental learning with large pre-trained models. As shown in Figure 1, FEDHAF is composed of three key components: (i) a feature extractor based on a frozen pre-trained model to generate high-quality representations, (ii) a feature adjustment

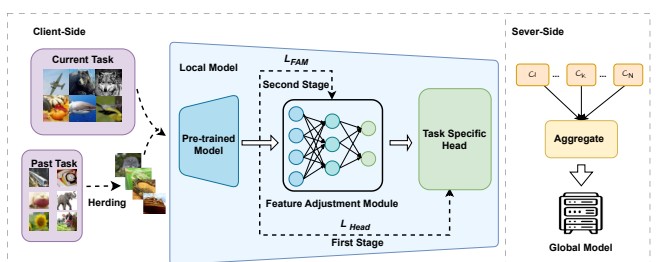

Figure 1: Overview of proposed FEDHAF.

module to align features across heterogeneous clients, and (iii) a task-specific head that adapts to new tasks while retaining useful prior knowledge. FEDHAF employs a two-stage training process. In the first stage, features are extracted using the frozen backbone. In the second stage, only the adjustment module and task-specific head are fine-tuned to incorporate new information while maintaining performance on previous tasks. Additionally, we introduce a feature-level consistency loss that encourages alignment between old and new representations, helping the global model preserve learned knowledge.

In this work, our contributions can be summarized as follows:

- We propose FEDHAF, a federated class-incremental learning framework that reduces catastrophic forgetting when using large pre-trained models.
- We design an efficient adaptation strategy that freezes the feature extractor and fine-tunes lightweight components, enabling low-cost and privacy-preserving model updates.
- We evaluate FEDHAF on benchmarks and show improved performance in both knowledge retention and class-imbalanced scenarios under heterogeneous distributions.

## 2 BACKGROUND AND MOTIVATION

### 2.1 FEDERATED LEARNING WITH PRE-TRAINED MODELS

Federated learning is a decentralized learning paradigm that allows a set of clients, denoted as $\mathcal{N} = \{1, \ldots, N\}$, to collaboratively train a global model without exposing their local data. A central server coordinates the training process by minimizing the following global objective:

$$\min_{\mathbf{w}} \mathcal{L}(\mathbf{w}, \mathcal{D}) = \sum_{i=1}^{N} \frac{|\mathcal{D}_i|}{|\mathcal{D}|} \mathcal{L}_i(\mathbf{w}, \mathcal{D}_i), \tag{1}$$

where $\mathbf{w}$ represents the global model parameters, $\mathcal{D}_i$ is the local dataset of client $i$, and $\mathcal{D} = \cup_{i=1}^{N} \mathcal{D}_i$ is the total data. Practical FL algorithms such as FedAvg (McMahan et al., 2017) and FedProx (Li et al., 2020) enable efficient training through local computation and periodic model aggregation. However, FL systems face two major challenges: (i) statistical heterogeneity caused by non-IID client data and (ii) communication constraints due to limited resources. These challenges lead to inconsistent local updates, which reduce both convergence speed and generalization performance.

To address these issues, recent works have explored incorporating large pre-trained models such as ResNet (He et al., 2016), BERT (Devlin et al., 2018), and GPT (Radford et al., 2019). These models

provide strong feature representations and typically require fewer training epochs to achieve high accuracy. For example, as shown in Figure 2, a pre-trained ResNet152 can reach over 80% accuracy on CIFAR-100 in just 10 epochs, while a randomly initialized model fails to surpass 52% accuracy even after 30 epochs. This accelerated convergence is particularly advantageous in federated settings with limited resources.

Despite their advantages, directly applying pre-trained models in federated learning introduces new challenges, particularly in dynamic or task-incremental scenarios. A major concern is *catastrophic forgetting*, where model updates from new tasks overwrite previously acquired knowledge. As illustrated in Figure 3, both FedAvg and FedProx experience significant accuracy degradation. The

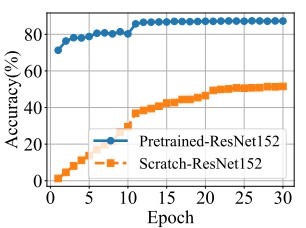 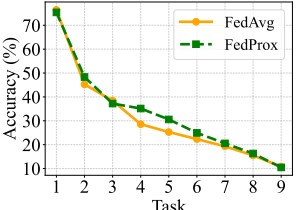

Figure 2: Model accuracy on CIFAR-100 dataset.

Figure 3: Forgetting issue in pre-trained models.

accuracy drops from over 75% on the first task to around 10% by the ninth task. FedAvg shows a rapid decline, reaching 45.25% by the second task. Although FedProx initially maintains a slightly higher accuracy of 48.33%, it eventually encounters a similar performance drop. This issue is further aggravated by the privacy constraints in FL, which prohibit access to past data for retraining or rehearsal. As a result, the global model struggles to preserve useful representations for new tasks.

## 2.2 EXISTING SOLUTIONS AND MOTIVATION

Catastrophic forgetting has been extensively studied in centralized class-incremental learning (CIL), where methods such as experience replay (Rolnick et al., 2019a), knowledge distillation (Li & Hoiem, 2017), and parameter isolation (Fernando, 2017) have shown effectiveness. However, these strategies often rely on accessing historical data or maintaining a global memory buffer, which conflicts with the privacy constraints of federated learning.

To adapt to federated settings, recent advances in Federated Class Incremental Learning (FCL) (Qi et al., 2023; Babakniya et al., 2023; Nori et al., 2025; Sun et al., 2024; Chaudhry et al., 2023) have proposed strategies such as client-side generative replay, server-side prototype synthesis, and data-free knowledge retention. While these methods can improve knowledge preservation, they often incur considerable computational and memory overhead, hindering deployment on resource-constrained devices. Furthermore, training generative models or synthetic exemplars under non-IID federated conditions is particularly challenging due to statistical heterogeneity and limited client capacity (Jeong & Moon, 2023). An alternative research direction focuses on modular adaptation of large pre-trained models. A widely adopted strategy is to freeze the backbone and fine-tune lightweight task-specific heads or adapters (Lyu & Liu, 2025; He et al., 2025; Lin et al., 2023), which effectively reduces both training cost and communication load. In parallel, data-free knowledge distillation methods (Yin et al., 2020; Chen et al., 2019; Zhao et al., 2023) have been explored to retain prior knowledge without requiring access to original data. However, these approaches often lack mechanisms to detect task shifts across clients, leading to degraded performance under dynamic task distributions (Zhou et al., 2024).

These limitations highlight the need for a unified, efficient framework that leverages the generalization of large pre-trained models, mitigates forgetting in dynamic federated settings, and supports scalability and privacy protection.

## 3 METHODOLOGY

We propose **FEDHAF**, as depicted in Figure 1, a modular framework that addresses the challenges of catastrophic forgetting, resource constraints, and task shifts in federated class-incremental learning. FEDHAF leverages the stability of large pre-trained models while introducing lightweight and privacy-preserving components to enable scalable deployment. Specifically, it consists of three modules: a frozen feature extractor $\phi$ that generates consistent representations across tasks, a Feature Adjustment Module (FAM) that aligns feature distributions using distillation and balance losses,

and a Task-Specific Head (TSH) trained to integrate new knowledge without overwriting previous decision boundaries. This design enables efficient, robust continual learning in federated settings.

## 3.1 DESIGN OF FEATURE ADJUSTMENT MODULE

FAM is designed to adapt fixed backbone features to new tasks in federated class-incremental learning. It is implemented as a multi-layer perceptron to transform frozen features into task-adaptive representations.In non-IID settings, newly introduced classes may shift feature distributions and undermine prior representations.

To mitigate this, FAM refines the intermediate features before classification to support task integration. Inspired by work in class imbalance (Guo & Zhang, 2017; Liu et al., 2017) and representation fairness in incremental learning (Zhu et al., 2022), we introduce two complementary objectives: *Feature Distillation Loss*, which constrains the drift of new features from old ones, and *Feature Balance Loss*, which aligns the feature magnitudes between new and old classes. This combination enables effective adaptation while preserving prior knowledge under frozen backbone constraints.

**Feature Distillation Loss**: Unlike traditional methods that distill information from the classifier, FEDHAF distills at the feature level. We leverage the old-class dataset $D^{\text{old}}$ to enforce consistency between the adjusted features produced by the current model $F_t$ and those generated by the previous task's model $F_{t-1}$. The loss is formally defined as:

$$L_{\text{FD}} = \frac{1}{|D^{\text{old}}|} \sum_{x \in D^{\text{old}}} \|F'_t(x) - F'_{t-1}(x)\|_2^2 \tag{2}$$

where $F'_t(x)$ and $F'_{t-1}(x)$ denote the adjusted feature representations of sample $x$ from the current and previous models, respectively. By aligning features in this way, the model is encouraged to maintain stable representations of old classes even as new tasks are introduced. This mechanism is critical in federated incremental learning, where revisiting past data is restricted by privacy concerns.

**Feature Balance Loss**: In incremental learning, the feature space for new tasks may dominate, causing the classifier to favor new tasks and neglect previously learned ones. To address this, FEDHAF introduces the *Feature Balance Loss*, which equalizes the feature norms across new and old tasks:

$$L_{\text{FB}} = \left| \frac{1}{C_{\text{new}}} \sum_{j=1}^{C_{\text{new}}} \|F_j\| - \frac{1}{C_{\text{old}}} \sum_{i=1}^{C_{\text{old}}} \|F_i\| \right| \tag{3}$$

where $C_{\text{new}}$ and $C_{\text{old}}$ represent the number of classes in the new and old tasks, respectively, and $\|F_j\|$ and $\|F_i\|$ denote the magnitudes of the feature vectors for samples from the new and old tasks. This loss ensures that the new task features do not overshadow the older task features, promoting a balanced learning process across all tasks.

In summary, the combination of *Feature Distillation Loss* and *Feature Balance Loss* allows FED-HAF to address catastrophic forgetting while maintaining adaptability. The final loss for FAM is given by: $L_{\text{FAM}} = \lambda_{FD} L_{\text{FD}} + \lambda_{FB} L_{\text{FB}}$, where $\lambda_{FD}$ and $\lambda_{FB}$ are hyperparameters that control the trade-off between feature consistency and feature balance. This ensures the seamless integration of new knowledge without compromising performance on older tasks.

## 3.2 DESIGN OF TASK-SPECIFIC HEAD

TSH classifies features adapted from FAM while accommodating new classes over time. It is implemented as a single-layer classifier with batch normalization, which improves training stability under evolving task distributions. To balance learning of new tasks with retention of previous knowledge, TSH is optimized using a combination of cross-entropy and knowledge distillation losses.

**Cross-Entropy Loss**: This CE loss function is employed to optimize the classification of new task samples. It encourages the classifier to learn accurate decision boundaries over the new task dataset $D^{\text{new}}$, where each pair $(x, y)$ denotes an input and its ground-truth label from classes introduced at task $t$. The loss is defined as:

$$L_{CE} = \frac{1}{|D^{\text{new}}|} \sum_{(x,y) \in D^{\text{new}}} CE(F_t(x), y), \tag{4}$$

where $F_t(x)$ represents the output logits of the current model $F_t$ for input $x$. Averaging over $D^{\text{new}}$ ensures that the classifier learns accurate decision boundaries for newly introduced classes.

**Knowledge Distillation Loss**: To mitigate catastrophic forgetting, we employ knowledge distillation to align the predictions of the current model $F_t$ with those of the previous task model $F_{t-1}$ on the old task dataset $D^{\text{old}}$. This encourages the current model to preserve decision boundaries learned from past tasks while adapting to new ones. The loss is defined as:

$$L_{KD} = \frac{1}{|D^{\text{old}}|} \sum_{x \in D^{\text{old}}} KL\big(F_t(x) \parallel F_{t-1}(x)\big), \tag{5}$$

where $F_t(x)$ and $F_{t-1}(x)$ denote the predictive distributions of the current and previous models for input $x$, respectively. By averaging over $D^{\text{old}}$, this formulation ensures that the knowledge encoded in earlier tasks is retained during continual updates.

The TSH loss is a weighted combination of the Cross-Entropy and Knowledge Distillation: $L_{\text{Head}} = \lambda_{\text{CE}}L_{\text{CE}} + \lambda_{\text{KD}}L_{\text{KD}}$, where $\lambda_{\text{CE}}$ and $\lambda_{\text{KD}}$ balance new class accuracy and old class retention, enabling the classifier to adapt to new tasks while preserving past performance.

### 3.3 FEDHAF TRAINING FRAMEWORK

Inspired by a recent study in incremental learning with large pre-trained models (Qin et al., 2023). FEDHAF adopts a two-stage training scheme that decouples task-specific adaptation from feature refinement. This modular approach ensures stable learning by preventing gradient interference between components and minimizing forgetting. Algorithm 1 illustrates different steps of FEDHAF.

**Two-Stage Training Process**. The training process of FEDHAF is structured into two stages: one focusing on the adaptation of the Task-Specific Head and the other on fine-tuning FAM.

During both stages, the feature extractor ($\phi$) remains fixed, ensuring that pre-trained feature representations are preserved. This design guarantees that the feature space remains generalizable across tasks and improves computational efficiency by reducing the number of parameters that need to be optimized. The key steps are as follows:

In the first stage, only the task-specific head is updated, while FAM remains frozen. The head is trained on new task data and herding exemplars from prior tasks. Cross-entropy loss guides learning on the current task, while knowledge distillation encourages retention of earlier decision boundaries. This stage enables rapid adaptation to new classes without altering the feature space.

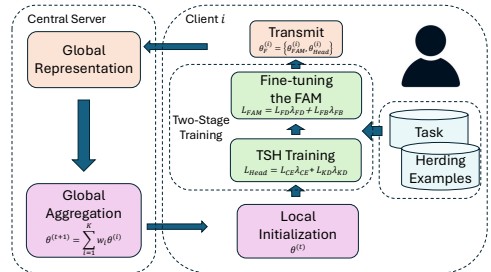

Figure 4: Two-stage training for FEDHAF.

Once the head is optimized, the model switches to refining FAM while keeping head fixed. This stage adjusts intermediate features to align new representations with those from previous tasks. Both feature distillation and feature balance losses are applied, ensuring integration of new knowledge while preserving consistency in the feature space. A small proportion of herding exemplars is introduced to mitigate feature drift and reduce catastrophic forgetting. Fine-tuning FAM in this manner ensures that the feature extractor adapts smoothly to newly introduced tasks while maintaining consistency with the knowledge acquired from previous tasks.

**Local Herding Sample Management**. To address forgetting without violating privacy constraints, each client maintains a small set of locally stored *herding exemplars* from previous tasks. These samples are never shared with the server or other clients. At the end of task $t$, each client selects a fixed number of representative samples per class from its current local dataset $\mathcal{D}_t^{(k)}$. Selection is based on feature-space proximity to the class mean using the frozen extractor:

$$x_c^{(i)} \in \arg\min_{x \in \mathcal{D}_c} \big\|\phi(x) - \bar{f}_c\big\|, \quad \bar{f}_c = \frac{1}{|\mathcal{D}_c|} \sum_{x \in \mathcal{D}_c} \phi(x) \tag{6}$$

where $\mathcal{D}_c \subseteq \mathcal{D}_t^{(k)}$ is the set of samples from class $c$. These exemplars form the local memory $\mathcal{D}_{\text{herd}}^{(k)}$, subject to a fixed buffer size $B$. During training for task $t+1$, clients use both $\mathcal{D}_{t+1}^{(k)}$ and $\mathcal{D}_{\text{herd}}^{(k)}$

to compute loss terms. This local replay mechanism enhances retention of prior knowledge while preserving privacy.

**Server Aggregation and Model Broadcasting.** Once local training is completed, each client sends its updated FAM and Task-Specific Head parameters to the server. The pre-trained feature extractor remains frozen and is excluded from communication. The server aggregates the received updates using standard federated averaging and forms the updated global model. These parameters are then broadcast back to all clients for the next incremental round. This process ensures low communication overhead, no raw data sharing, and full compliance with federated privacy constraints.

### 3.4 THEORETICAL ANALYSIS

To support the empirical results of FEDHAF, we provide theoretical guarantees under standard assumptions in federated optimization. Let $\theta = (\theta_{\text{Head}}, \theta_{\text{FAM}})$ denote the learnable parameters of FEDHAF. For simplicity, let each client minimize the following local objective:

$$L_{\text{loc}}(\theta) = \lambda_{\text{CE}}L_{\text{CE}} + \lambda_{\text{KD}}L_{\text{KD}} + \lambda_{\text{FD}}L_{\text{FD}} + \lambda_{\text{FB}}L_{\text{FB}}, \tag{7}$$

where each loss term corresponds to cross-entropy, knowledge distillation, feature-level distillation, and feature balance regularization, respectively. We now introduce the used assumptions (Wang et al., 2024; Lian et al., 2018):

**Assumption 1** (Lipschitz Smoothness). Loss function $L_{\text{loc}}$ is differentiable and has $L$-Lipschitz continuous gradients:

$$\|\nabla L_{\text{loc}}(\theta) - \nabla L_{\text{loc}}(\theta')\| \leq L\|\theta - \theta'\|, \quad \forall \theta, \theta'. \tag{8}$$

**Assumption 2** (Bounded Gradient Variance). Let $\xi$ denote a random minibatch. Variance of the stochastic gradient is:

$$\mathbb{E}_{\xi} \|\nabla L_{\text{loc}}(\theta; \xi) - \nabla L_{\text{loc}}(\theta)\|^2 \leq \sigma^2. \tag{9}$$

First, we introduce the convergence result under the federated averaging scheme.

**Theorem 1** (Convergence). *Under Assumptions 1 and 2, suppose each client performs $E$ local updates with learning rate $\eta$ satisfying $\eta L E \leq 1$. Then after $T$ global rounds, the expected average gradient norm satisfies:*

$$\frac{1}{T} \sum_{t=0}^{T-1} \mathbb{E}\left[\left\|\nabla L_{\text{loc}}(\theta^t)\right\|^2\right] \leq \frac{2(L_{\text{loc}}(\theta^0) - L^*)}{\eta E T} + \frac{\eta L \sigma^2 (E-1)}{2K}, \tag{10}$$

*where $L^*$ is the optimal loss and $K$ is the number of clients.*

Next, we analyze how changes in adjusted features contribute to forgetting.

**Theorem 2** (Bound on Catastrophic Forgetting). *Let $F'_{\text{old}}$ and $F'_{\text{new}}$ denote the adjusted features before and after the update, respectively. If*

$$\mathbb{E}\left[\|F'_{\text{new}} - F'_{\text{old}}\|^2\right] \leq \varepsilon, \tag{11}$$

*and $h$ is $L_h$-Lipschitz, then the increase in classification error on old tasks satisfies:*

$$\Delta E_{\text{old}} \leq L_h \sqrt{\varepsilon}. \tag{12}$$

**Theorem 3** (Stability of Two-Stage Training). *Assume that after Stage 1, $\|\nabla_{\theta_{\text{Head}}} L_{\text{loc}}\| \leq \epsilon_1$, and after Stage 2, $\|\nabla_{\theta_{\text{FAM}}} L_{\text{loc}}\| \leq \epsilon_2$. The full gradient norm is bounded as:*

$$\|\nabla L_{\text{loc}}(\theta)\| \leq \epsilon_1 + \epsilon_2. \tag{13}$$

This result shows that decoupling the modules reduces gradient interference and improves stability.

**Theorem 4** (Generalization Bound). *Let $\mathcal{H}$ denote the hypothesis class formed by the Feature Adjustment Module and classifier head. If the loss is bounded by $M$, then with probability at least $1 - \delta$, the generalization error satisfies:*

$$\mathbb{E}_{\text{test}} \leq \hat{E}_{\text{train}} + 2R_n(\mathcal{H}) + \sqrt{\frac{\log(1/\delta)}{2n}} + \mathcal{O}(\lambda_{\text{FB}}), \tag{14}$$

*where $R_n(\mathcal{H})$ is the empirical Rademacher complexity.*

These analyses demonstrate that the modular and decoupled architecture of FEDHAF ensures convergence, mitigates forgetting, and provides generalization guarantees in federated settings with non-IID and evolving tasks. Related proofs for all theoretical results are provided in Appendix E.

## 4 EXPERIMENTS

### 4.1 EXPERIMENT SETTINGS

**Datasets and Models.** We evaluate FEDHAF on four benchmarks covering both vision and language domains: CIFAR-100 (Krizhevsky, 2009), TinyImageNet (Le & Yang, 2015), ImageNet (Deng et al., 2009), and Shakespeare (Caldas et al., 2018). CIFAR-100 contains 100 classes with 600 images per class, while TinyImageNet includes 200 classes with 500 training and 50 validation images each. ImageNet provides a large-scale visual recognition benchmark with 1,000 classes and over 1.2 million training images, posing significant challenges for scalability and generalization. Shakespeare is a character-level language modeling dataset partitioned by speaker, characterized by naturally imbalanced and variable-length samples. For the image classification tasks, we use ResNet-152 as the backbone model. For the Shakespeare dataset, we adopt a pre-trained BERT-BASE to capture sequence-level linguistic patterns.

**Parameter Settings.** We adopt the class-incremental learning setup, where new classes are introduced progressively over time. To simulate realistic federated environments, we generate non-IID data partitions using Latent Dirichlet Allocation (LDA) following the approach in (Babakniya et al., 2023), resulting in heterogeneous and imbalanced data distributions (see Table 1). All models are trained using stochastic gradient descent (SGD) with a learning rate of 0.02. To reduce forgetting, we apply a herding strategy that retains 20% of samples from previous tasks, and this ratio is varied in the ablation studies. The settings for LANDER follow those described in (Tran et al., 2024b).

**Evaluation Metrics.** To assess the performance of FEDHAF in federated class-incremental learning, we adopt three standard metrics: average accuracy, average forgetting, and wallclock time. *Average Accuracy* ($\tilde{A}$) measures overall classification

Table 1: Training parameters of each benchmark dataset.

| Dataset | #Client | #Client per round | #classes per task |
|---|---|---|---|
| CIFAR-100 | 50 | 5 | 10 |
| TinyImageNet | 50 | 5 | 20 |
| ImageNet | 300 | 30 | 5 |
| Shakespeare | 100 | 10 | 6 |

performance by computing the test accuracy over all seen classes at the end of each incremental round. The final value is obtained by averaging across all rounds, reflecting the model's ability to retain and integrate knowledge throughout the learning process. *Average Forgetting* ($\tilde{F}$) quantifies the degradation in performance on previously learned tasks. It is calculated as the difference between the maximum accuracy achieved during training and the final accuracy on each task, then averaged over all tasks to capture the model's retention capability. *Wallclock Time* reports the average time required to complete one federated round, measured in seconds using an NVIDIA RTX 3090 GPU. This value is averaged across clients to evaluate training efficiency under practical conditions.

**Baselines.** We compare FEDHAF against six methods, including FedAvg (McMahan et al., 2017), FedProx (Li et al., 2020), FedCIL (Qi et al., 2023), FedLwF-2T (Usmanova et al., 2021), LANDER (Tran et al., 2024b), and Oracle. FedAvg performs simple parameter averaging, while FedProx stabilizes training under heterogeneous data via a proximal term. FedCIL leverages prototype alignment and classifier regularization to alleviate forgetting in class-incremental scenarios. FedLwF-2T adapts the learning-without-forgetting paradigm to federated settings via dual-temperature knowledge distillation. LANDER addresses data-free continual learning by synthesizing samples around label-text anchors to preserve prior knowledge. Finally, Oracle assumes centralized access to all data across clients and tasks, thus providing an upper-bound reference for performance.

### 4.2 EXPERIMENTAL RESULTS

**Main Results.** Figure 5 illustrates the accuracy trajectories of FEDHAF and baseline methods on CIFAR-100, TinyImageNet, ImageNet, and Shakespeare. As tasks accumulate, performance degradation occurs for all methods. However, FEDHAF outperforms baseline approaches across datasets. Specifically, on CIFAR-100, FEDHAF maintains superior accuracy throughout incremental learning, significantly surpassing FedAvg and FedProx, which rapidly decline after a few tasks. LANDER demonstrates relatively stable performance, while the Oracle model maintains stable high performance, emphasizing the importance of comprehensive data availability.

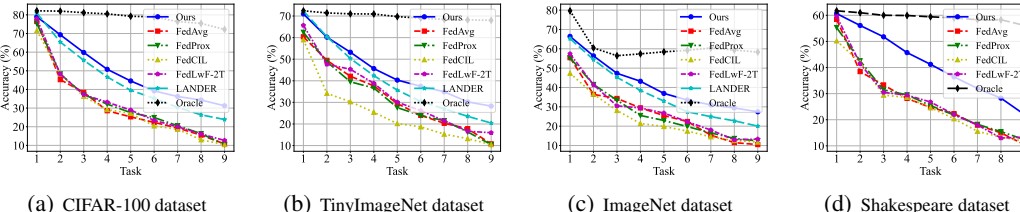

(a) CIFAR-100 dataset     (b) TinyImageNet dataset     (c) ImageNet dataset     (d) Shakespeare dataset

Figure 5: Accuracy vs. tasks. After each task, the model is evaluated on all previously seen tasks.

Table 2: Ablation study on CIFAR-100 evaluating the impact of removing key components.

| Method | $A^1$ | $A^2$ | $A^3$ | $A^4$ | $A^5$ | $A^6$ | $A^7$ | $A^8$ | $A^9$ | $\tilde{A} \uparrow$ | $\tilde{F} \downarrow$ |
|---|---|---|---|---|---|---|---|---|---|---|---|
| Ours - w/o $L_{\text{CE}}$ | 46.2 | 31.17 | 23.28 | 18.66 | 15.43 | 13.19 | 11.47 | 10.26 | 9.17 | 19.87 | 58.8 |
| Ours - w/o $L_{\text{KD}}$ | 65.43 | 53.4 | 44.37 | 39.21 | 35.32 | 28.94 | 20.99 | 17.32 | 12.56 | 35.28 | 43.38 |
| Ours - w/o $L_{\text{FD}}$ | 69.71 | 63.39 | 53.37 | 45.32 | 39.11 | 32.1 | 28.94 | 25.96 | 22.43 | 42.93 | 36.41 |
| Ours - w/o $L_{\text{FB}}$ | 62.85 | 52.47 | 45.66 | 39.67 | 35.39 | 29.41 | 25.32 | 19.32 | 20.67 | 36.75 | 41.91 |
| Ours - w/o FAM | 60.32 | 49.3 | 41.56 | 32.59 | 25.67 | 23.22 | 21.09 | 18.47 | 15.99 | 32.02 | 46.64 |
| Ours - w/o 2-stages training | 71.69 | 64.42 | 55.69 | 47.22 | 39.16 | 35.69 | 32.15 | 29.32 | 25.12 | 43.49 | 35.17 |
| Ours | 79.01 | 69.32 | 59.84 | 50.81 | 44.55 | 39.22 | 36.21 | 34.32 | 31.21 | 49.39 | 30.3 |

Table 3 further quantifies these results. FEDHAF reaches 49.39% average accuracy with only 30.03% forgetting, compared to over 60% forgetting for FedAvg and FedProx. FedLwF-2T achieves modest improvements through distillation but remains weak across tasks. The *Oracle* model provides an upper bound at 78.67%. LANDER is more robust than FedAvg and FedProx but still trails FEDHAF in both accuracy and retention. Similar patterns are observed on TinyImageNet and ImageNet, where FEDHAF maintains nearly 42% average accuracy on TinyImageNet and demonstrates strong stability on ImageNet despite increased task complexity and data volume. On the Shakespeare dataset, which poses unique challenges due to its sequential and linguistic nature, most baselines suffer severe forgetting, while FEDHAF maintains over 20% accuracy on the final task. Since LANDER relies on pseudo feature generation and visual class-center structures, its assumptions do not hold for natural language tasks; for fairness, we omit it from the Shakespeare experiments. These results further confirm that FEDHAF offers consistency across both vision and language tasks in federated class-incremental settings.

In addition to accuracy and forgetting, Table 3 also reports training time. FEDHAF requires approximately 67.3 seconds per task, which is higher than FedAvg and FedProx due to their use of simple local updates without explicit knowledge retention mechanisms. FedLwF-2T introduces a distillation objective with moderate computational overhead, while FedCIL takes

Table 3: Performance of baselines in terms of average accuracy and average forgetting on the CIFAR-100 dataset.

| Method | Avg. Accuracy | Avg. Forgetting | Training Time (s) |
|---|---|---|---|
| FedAvg (McMahan et al., 2017) | 31.32% ± 0.11 | 65.68% ± 0.91 | ≈16.5 |
| FedProx (Li et al., 2020) | 33.21% ± 0.32 | 64.93% ± 0.48 | ≈19.4 |
| FedCIL (Qi et al., 2023) | 34.55% ± 0.48 | 61.14% ± 0.42 | ≈32.1 |
| FedLwF-2T (Usmanova et al., 2021) | 33.53% ± 0.19 | 65.26% ± 0.82 | ≈18.3 |
| LANDER (Tran et al., 2024b) | 40.09% ± 0.52 | 35.26% ± 0.77 | ≈198.4 |
| FEDHAF (Ours) | 49.39% ± 0.23 | 30.3% ± 0.59 | ≈67.3 |
| Oracle | 78.67% ± 0.33 | – | ≈132.6 |

about 32.1 seconds per task because of the added cost of training generative models. LANDER incurs longer training time mainly due to its adversarial training between generator and discriminator, along with the overhead of generating and distilling a large number of pseudo features in each task round. In summary, FEDHAF achieves significantly better performance. Its modular design, which incorporates representation preservation and task-aware adaptation, enhances knowledge retention while achieving a favorable trade-off between efficiency and stability.

Overall, FEDHAF demonstrates substantial advantages in mitigating catastrophic forgetting in federated class-incremental learning across diverse visual and language tasks. Its structured approach to knowledge retention and adaptation ensures consistent high accuracy and manageable training complexity, highlighting its practicality and scalability in realistic federated learning scenarios.

**Evaluation on Loss Functions and Architectural Components.** We perform additional experiments on CIFAR-100 under the federated incremental learning setting to assess the contribution of each loss function and architectural component. Table 2 summarizes the results. Disabling the cross-entropy loss corresponds to using all classifier heads for supervision instead of restricting updates to task-relevant outputs. This introduces conflicting gradients from unseen classes and accelerates forgetting. In contrast, head splitting provides isolated supervision, stabilizing class boundaries. Re-

moving the knowledge distillation loss weakens retention by eliminating soft constraints that help regularize predictions across tasks. This leads to output drift, especially under non-IID aggregation. Feature-level distillation is critical for maintaining stable representations. Without it, feature embeddings shift unpredictably, erasing prior structure, a problem worsened in federated settings. Omitting the feature balance loss disrupts inter-task consistency, allowing new-task gradients to dominate and bias the feature space, which degrades performance on previous classes. Removing the FAM prevents alignment between frozen features and client-specific distributions, resulting in growing misalignment and unstable updates. Lastly, disabling two-stage training leads to gradient interference between classifier and feature adaptation, hindering convergence. Sequential optimization of these components ensures smoother transitions and better stability across tasks.

**Communication Cost.** FEDHAF minimizes communication by freezing the backbone and only transmitting the Feature Adjustment Module and Task-Specific Head. In contrast, LANDER transmits larger components, including task-specific latent vectors, multi-level prototypes, and generator parameters. These high-dimensional elements significantly raise communication costs. Prior work (Shin et al., 2017) highlights the overhead of generative replay in bandwidth-limited federated settings. Other baselines like FedCIL or FedLwF involve transferring full model outputs or multiple classifier copies, further increasing per-round costs. We further provide a detailed computational analysis of FEDHAF in Appendix D. FEDHAF makes it more scalable and deployment-friendly in resource-constrained environments.

**Impact of Herding Ratio.** We conduct an ablation study to evaluate the effect of the herding ratio ($\rho$), which denotes the proportion of retained data from previous tasks, on both model performance and training efficiency using the CIFAR-100 dataset. As shown in Table 4, increasing $\rho$ improves accuracy but also leads to higher computational cost. At $\rho = 100\%$, the model achieves the highest accuracy but re-

Table 4: Impact of herding ratio on CIFAR-100.

| Herding Ratio | Average Accuracy | Training Time |
|---|---|---|
| 100% | 51.94% | 50 mins |
| 75% | 51.49% | 38 mins |
| 50% | 50.87% | 29 mins |
| 40% | 50.35% | 21 mins |
| 30% | 49.98% | 16 mins |
| 20% | 49.39% | 13 mins |
| 10% | 43.23% | 11 mins |
| 5% | 36.98% | 9 mins |
| 1% | 30.70% | 8 mins |

quires 50 minutes of training. Reducing $\rho$ to 75% or 50% slightly decreases accuracy while substantially lowering training time, indicating a favorable trade-off between performance and efficiency. However, when $\rho$ falls below 30%, the accuracy drops sharply despite further reductions in computation. These results suggest that moderate herding ratios provide a good balance, whereas very low values impair learning stability and knowledge retention.

**Hyperparameter Tuning for FED-HAF.** Table 5 shows the effect of varying key hyperparameters on the final average accuracy $\tilde{A}$. The weights $\lambda_{FD}$, $\lambda_{FB}$, $\lambda_{CE}$, and $\lambda_{KD}$ control the contributions of feature

Table 5: Impact of different settings on CIFAR-100.

| $\lambda_{FD}$ | $\tilde{A}$ | $\lambda_{FB}$ | $\tilde{A}$ | $\lambda_{CE}$ | $\tilde{A}$ | $\lambda_{KD}$ | $\tilde{A}$ |
|---|---|---|---|---|---|---|---|
| 0.1 | 38.99 | 0.1 | 38.23 | 0.1 | 38.32 | 0.1 | 32.14 |
| 0.5 | 40.21 | 1 | 41.67 | 0.5 | 40.31 | 0.5 | 36.22 |
| 1 | 41.09 | 10 | 39.43 | 1 | 42.67 | 1 | 42.67 |
| 2 | 43.67 | 50 | 38.32 | 2 | 29.42 | 2 | 40.1 |

distillation, feature balance, cross-entropy, and knowledge distillation losses, respectively. Proper tuning of these values is essential for balancing feature alignment, classification performance, and knowledge retention. Based on empirical results, we set $\lambda_{FD} = 2$, $\lambda_{FB} = 1$, $\lambda_{CE} = 1$, and $\lambda_{KD} = 1$ as the default configuration in all experiments.

## 5 CONCLUSIONS AND LIMITATIONS

We propose **FEDHAF**, a modular framework for federated class-incremental learning with large pre-trained models. FEDHAF freezes the feature extractor and decouples feature alignment from classifier adaptation, enabling efficient and stable local training. It employs a two-stage training strategy with feature-level distillation and balance regularization to effectively retain knowledge across tasks and reduce forgetting. This design ensures reasonable communication and computation overhead, making FEDHAF suitable for privacy-sensitive, non-IID federated settings. Extensive experiments show that FEDHAF consistently outperforms strong baselines in both accuracy and forgetting. However, the current design assumes synchronized updates and task-specific heads, which may limit flexibility in asynchronous or overlapping class scenarios. Future work will further explore dynamic head sharing, asynchronous training, and integration with formal privacy guarantees.

ETHICS STATEMENT

All authors have read and adhere to the ICLR Code of Ethics.

REPRODUCIBILITY STATEMENT

We provide implementation details in the Experiments and Appendix, including network architectures, hyperparameters, and environment configurations. All experiments are conducted using a fixed random seed of 1234 to ensure reproducibility. If the paper is accepted, we will release the code on GitHub. If the reviewers need to check the code during the review process or in the rebuttal stage, we will upload it to an anonymous GitHub repository.

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

## A  THE USE OF LARGE LANGUAGE MODELS (LLMS)

Large language models were used solely to aid in polishing the writing of this paper. They were not used for research ideation, methodology, analysis, or drawing conclusions. The authors take full responsibility for all content.

## B  FEDHAF INCREMENTAL TRAINING ALGORITHM

Algorithm 1 summarizes our method.

---

**Algorithm 1** FEDHAF Incremental Training Algorithm

---

**Require:** Initial model parameters $\theta_{\text{FAM}}, \theta_{\text{Head}}$, new task data $D_{\text{new}}$, herding exemplars from old tasks $D_{\text{herding}}$, previous task knowledge.
**Ensure:** Updated model parameters $\theta_{\text{FAM}}, \theta_{\text{Head}}$
 1: **Stage 1: Train Task-Specific Head (Freeze FAM)**
 2: **for** each epoch in Stage 1 **do**
 3:     **for** each minibatch $(X, Y)$ in $D_{\text{new}} \cup D_{\text{herding}}$ **do**
 4:         Compute logits $p = \text{Head}(\text{FAM}(X))$
 5:         Compute classification and distillation losses
 6:         Backpropagate and update $\theta_{\text{Head}}$ (FAM remains frozen)
 7:     **end for**
 8: **end for**
 9: **Stage 2: Fine-tune FAM (Freeze Head)**
10: **for** each epoch in Stage 2 **do**
11:     **for** each minibatch $(X, Y)$ in $D_{\text{new}} \cup D_{\text{herding}}$ **do**
12:         Compute adjusted features $F'_{\text{new}} = \text{FAM}(X)$
13:         Compute feature alignment loss
14:         Backpropagate and update $\theta_{\text{FAM}}$ (Head remains frozen)
15:     **end for**
16: **end for**
17: **Server Aggregation**
18: Clients send $\theta_{\text{FAM}}, \theta_{\text{Head}}$ to the server
19: Server aggregates client models: $\theta_{\text{global}} = \sum_{k=1}^{N} w_k \theta_k$
20: Server redistributes the global model to clients **return** Updated global model $\theta_{\text{global}}$

---

## C    DETAILS OF THE FEDHAF

**Architectures of FEDHAF.**

Table 6: Details for FEDHAF Architecture

| FeatureAdjustment | ClassificationHead |
|---|---|
| Linear(2048 → 512) | Linear(256 → 128) |
| BatchNorm1d(512) | BatchNorm1d(128) |
| ReLU | ReLU |
| Dropout | Dropout |
| Linear(512 → 512) | Linear(128 → num_classes) |
| BatchNorm1d(1024) | |
| ReLU | |
| Dropout | |
| Linear(512 → 256) | |
| BatchNorm1d(128) | |
| Residual Add | |

**Weight Initialization.** For image classification tasks (CIFAR-100, TinyImageNet, ImageNet), we adopt Kaiming initialization for all trainable modules at each new task. This initialization is well-suited to convolutional layers and accelerates convergence by preserving variance across layers. In contrast, for text-based tasks (e.g., Shakespeare), the model is randomly initialized.

**Global Aggregation Method.** Once local training for task $t$ is complete, each client uploads its updated parameters $(\theta_{\text{FAM}}^{(k,t)}, \theta_{\text{Head}}^{(k,t)})$ to the server. The shared feature extractor $\phi$ remains fixed and is excluded from communication. The server performs weighted Federated Averaging (FedAvg) to aggregate the received updates:

$$\theta_{\text{FAM}}^{(t)} = \sum_{k=1}^{K} w_k \theta_{\text{FAM}}^{(k,t)}, \quad \theta_{\text{Head}}^{(t)} = \sum_{k=1}^{K} w_k \theta_{\text{Head}}^{(k,t)} \tag{15}$$

where $w_k = \frac{|\mathcal{D}_t^{(k)}|}{\sum_j |\mathcal{D}_t^{(j)}|}$ reflects the relative data volume of each client. The updated global parameters $(\theta_{\text{FAM}}^{(t)}, \theta_{\text{Head}}^{(t)})$ are then broadcast to all clients before the next task round begins. As only model parameters are exchanged, FEDHAF ensures low communication cost and strong privacy guarantees.

# D  COMPLEXITY ANALYSIS

The computational complexity of FEDHAF is evaluated by analyzing the time complexity of each stage of the training process.

In Stage 1, the computational complexity is driven by the number of minibatches, the dimensionality of the Task-Specific Head, and the operations involved in forward and backward propagation. The complexity per iteration in this stage is expressed as:

$$O(N_{\text{minibatch}} \times (d_{\text{head}} + N_{\text{exemplar}})), \tag{16}$$

where $N_{\text{minibatch}}$ is the number of minibatches processed per epoch, $d_{\text{head}}$ represents the dimensionality of the Task-Specific Head, and $N_{\text{exemplar}}$ is the number of exemplars used for knowledge distillation. Consequently, the total time complexity for Stage 1 is:

$$O(E_1 \times N_{\text{minibatch}} \times (d_{\text{head}} + N_{\text{exemplar}})), \tag{17}$$

where $E_1$ denotes the number of epochs in this stage.

Stage 2 introduces complexity due to the fine-tuning of the FAM. The complexity per iteration in this stage is:

$$O(N_{\text{minibatch}} \times (d_{\text{FAM}} + N_{\text{exemplar}})), \tag{18}$$

where $d_{\text{FAM}}$ represents the dimensionality of the Feature Adjustment Module. Therefore, the total time complexity for Stage 2 is:

$$O(E_2 \times N_{\text{minibatch}} \times (d_{\text{FAM}} + N_{\text{exemplar}})), \tag{19}$$

with $E_2$ indicating the number of epochs in Stage 2. In the context of federated learning, each client performs local training on their data, and the server aggregates the models using a weighted averaging strategy. The complexity of the aggregation step is given by:

$$O(N_{\text{clients}} \times d_{\text{global}}), \tag{20}$$

where $N_{\text{clients}}$ is the number of participating clients in the federated learning process, and $d_{\text{global}}$ is the global model's dimensionality. This aggregation step is performed after each local training round.

Additionally, the communication cost for transmitting model updates between the clients and the server is influenced by both the number of clients and the size of the model. The communication complexity is given by:

$$O(N_{\text{clients}} \times d_{\text{global}}), \tag{21}$$

which accounts for the transfer of the model parameters from each client to the server and the subsequent redistribution of the aggregated global model back to the clients.

These characteristics indicate that FEDHAF achieves a favorable balance between learning effectiveness and computational efficiency. By freezing the backbone and restricting optimization to lightweight modules—namely the task-specific head and the Feature Adjustment Module (FAM)—the number of trainable parameters per client is reduced by over 95% compared to full model fine-tuning. For example, in our CIFAR-100 experiments using a ResNet-152 backbone, the FAM and head together comprise fewer than 2 million parameters, in contrast to over 60 million in the full network.

This modular and decoupled design not only reduces computational and memory demands, but also facilitates deployment on edge clients with limited resources, ensuring stable learning without compromising scalability or privacy guarantees.

# E   THEORETICAL ANALYSIS OF FEDHAF

## E.1   PROOF FOR THEOREM 1

*Proof.* Standard in literature on local SGD. Using the $L$-smoothness of $\mathcal{L}_{\text{loc}}$ and convexity of client averaging, one applies the descent lemma and bounds the error from local steps via Assumption 2. $\square$

## E.2   PROOF FOR THEOREM 2

*Proof.* For any feature vector $r$, let $p_{\text{old}} = h(r)$ and $p_{\text{new}} = h(r + \Delta r)$. By Assumption 3,

$$\|p_{\text{new}} - p_{\text{old}}\| \le L_h \|\Delta r\|.$$

Taking expectation over feature drift and applying Jensen's inequality gives

$$\mathbb{E}\|p_{\text{new}} - p_{\text{old}}\| \le L_h \sqrt{\mathbb{E}\|\Delta r\|^2} \le L_h \sqrt{\epsilon},$$

which bounds the shift in output logits, hence the increase in classification error. $\square$

## E.3   PROOF FOR THEOREM 3

*Proof.* We follow the standard Rademacher complexity framework. Since the total loss includes a regularized term $\mathcal{L}_{\text{FB}}$ that is Lipschitz in the features, its effect is upper bounded by $\lambda_{\text{FB}} L_\phi$ for some $L_\phi > 0$. Applying symmetrization and contraction bounds yields the result. $\square$

## E.4   PROOF FOR THEOREM 4

*Proof.* By block-separability of parameters, we write:

$$\nabla \mathcal{L}_{\text{loc}}(\theta) = \begin{bmatrix} \nabla_{\theta_{\text{Head}}} \mathcal{L}_{\text{loc}} \\ \nabla_{\theta_{\text{FAM}}} \mathcal{L}_{\text{loc}} \end{bmatrix}.$$

Then,

$$\|\nabla \mathcal{L}_{\text{loc}}(\theta)\| \le \|\nabla_{\theta_{\text{Head}}} \mathcal{L}_{\text{loc}}\| + \|\nabla_{\theta_{\text{FAM}}} \mathcal{L}_{\text{loc}}\| \le \varepsilon_1 + \varepsilon_2,$$

by triangle inequality in Euclidean norm. $\square$

