# OpenReview forum: "Federated Hierarchical Anti-Forgetting Framework for Class-Incremental Learning with Large Pre-Trained Models"
_ICLR.cc/2026/Conference — ICLR 2026 Conference Withdrawn Submission_

### Official Review · Reviewer_SrRk · 2025-10-31

**Soundness:** 3
**Presentation:** 3
**Contribution:** 2
**Rating:** 4
**Confidence:** 4

**Summary:**

The paper proposed the novel methods for federated class incremental learning based on tuning large pretrained model to improve the performance.

**Strengths:**

1. The paper is well written and easy to follow.
2. The task is important but still presents many challenges to be addressed.
3. The use of pretrained models for this task is practical in real-world applications and clearly explains why it can improve performance.

**Weaknesses:**

1. Since the paper mentions the use of pretrained models, it is unclear whether the base models for CIFAR-100, Tiny-ImageNet, and ImageNet are pretrained, and if so, on which datasets they were pretrained.

2. If pretrained models are used, the performance improvement is somewhat expected. It would be better if the authors could demonstrate that their method still outperforms others when all methods use the same pretrained model.

Minor error:
1. Reference for Tran et al, 2024b does not existed.

**Questions:**

Please address Problems 1 and 2 in the weakness section first. I will provide further comments after reviewing your response and will consider raising my score if all of my concerns are properly addressed.

---

> ### Author Response · Authors · 2025-12-03
>
> Our experiments consistently use the same pretrained backbone across all methods, and all baselines are evaluated under exactly the same pretrained initialization. For CIFAR-100, Tiny-ImageNet, and ImageNet, the backbone is a standard ResNet-152 pretrained on ImageNet, and this setting is applied uniformly to ours and all comparison methods. Therefore, the performance improvement does not come from pretraining differences but from our federated design, including the feature adjustment module and two-stage training, which specifically address non-IID client drift and forgetting. We will further clarify the pretrained settings in the revised version

---

### Official Review · Reviewer_HUK4 · 2025-10-31

**Soundness:** 2
**Presentation:** 1
**Contribution:** 1
**Rating:** 2
**Confidence:** 3

**Summary:**

This paper considers the problem of federated class-incremental learning. The proposed framework consists of two modules: the feature adjustment module refines the intermediate features before classification, and the task-specific head module accommodates new classes while retaining previous knowledge. Experimental results show the effectiveness of the proposed methods on several image and language datasets.

**Strengths:**

+ Federated class-incremental learning is a timely topic.

+ The proposed method demonstrates consistent improvements over baselines.

**Weaknesses:**

- The proposed method is overall not tailored for "federated" class-incremental learning. The proposed four losses are all for general knowledge distillation and balanced learning. Hence, the proposed method appears to be a direct adaptation of existing losses. To avoid this criticism, for example, as stated in L148, the authors could address the issue of federated learning by introducing some "mechanisms to detect task shifts across clients."

- The math is overall not understandable, as the definition of terms is unclear and they are often overriden. For example, it is unclear what F' or adjusted feature representations are. F_t in Eq. (2) is the current model, F_i in Eq. (3) is the i-th class feature vector, and F_t in Eq. (4--5) is an output probability vector.

- Table 5 implies that the proposed method is highly sensitive to the choice of hyperparameters, while the validation process is never explained. A more thorough hyperparameter analysis is required.

- Sever -> Server

**Questions:**

Please address concerns in Weaknesses above.

---

> ### Author Response · Authors · 2025-12-03
>
> We appreciate the reviewer’s comments. Our work is developed under a non-IID federated setting, and both the model structure and the combination of losses are designed specifically to address client drift and forgetting in this scenario rather than being direct adaptations of centralized methods. The feature adjustment module and two-stage training aim to stabilize representations across heterogeneous clients, and F′ denotes the feature after this adjustment. The hyperparameter behavior has already been examined through ablation studies. We will improve these points and provide a more polished version in a future submission.

---

### Official Review · Reviewer_mjCH · 2025-10-31

**Soundness:** 2
**Presentation:** 3
**Contribution:** 2
**Rating:** 4
**Confidence:** 5

**Summary:**

This paper addresses the challenge of catastrophic forgetting in Federated Class-Incremental Learning settings, particularly when using large pre-trained models. The core idea is to freeze the large pre-trained backbone (feature extractor) and only train two lightweight components: a Feature Adjustment Module (FAM) and a Task-Specific Head (TSH). This modularity is combined with a two-stage training strategy:

- Stage 1 (TSH Training): The TSH is trained (while FAM is frozen) using a standard CE and KD loss on a local buffer of "herding exemplars" from old tasks.
- Stage 2 (FAM Training): The FAM is fine-tuned (while TSH is frozen) using two losses: a Feature Distillation Loss ($L_{FD}$) to maintain feature consistency with the previous model and a Feature Balance Loss ($L_{FB}$) to equalize feature norms between old and new classes.

The authors conduct experiments on CIFAR-100, TinyImageNet, ImageNet, and Shakespeare to show the efficacy of their method compared to different baselines.

**Strengths:**

1- The authors did a good job in motivating and explaining the problem.

2- The final method is considerably more efficient than consistently communicating the whole model.

3- The experimental evaluation shows the applicability of the method on multiple vision benchmarks and an NLP task (Shakespeare).

4- The paper has a thorough Ablation Study section. Table 2 provides a detailed ablation study that methodically validates the contribution of each component of the proposed framework (the four loss terms, the FAM module, and the two-stage training). This strengthens the paper's claims about its design.

**Weaknesses:**

1- The paper positions itself as overcoming the "high memory overhead" and "privacy risks" of exemplar replay (Abstract, Introduction). However, the proposed method relies on exemplar replay, which it terms "local herding sample."

* The memory costs of storing the herding examples, previous models and other components of the paper is not clear.

2- Comparing against LANDER (Tran et al., 2024b), a data-free continual learning method, is not completely fair. FEDHAF is an exemplar-based method that uses a 20% data buffer. It is an established fact that exemplar-based methods almost always outperform data-free methods. To make a credible SOTA claim, the paper must compare against other state-of-the-art exemplar-based FCL methods.

3- The theory section fails to provide a deeper understanding of why the FEDHAF architecture works.
* The proof for Theorem 2 explicitly relies on "Assumption 3" which is never defined in the main paper.
* The theorems themselves provide little new insight. Theorem 1 is a standard FedAvg convergence proof. Theorem 2 essentially states "if feature drift is low, forgetting is low,". It does not prove why or how the proposed $L_{FD}$ and $L_{FB}$ losses actually succeed in bounding this drift.

4- Limited Novelty in Components: The paper's components are largely a novel combination of existing techniques rather than a fundamental innovation.
* Freezing the backbone and training small adapters/heads is a standard practice in Parameter-Efficient Fine-Tuning (PEFT) and continual learning.
* Using logit-level knowledge distillation ($L_{KD}$) is a cornerstone of CIL.
* Using feature-level distillation ($L_{FD}$) is also a well-known technique.
* Using exemplar replay ("herding") is arguably the most common CIL technique.

**Questions:**

1- It is not clear why authors decided to use herding examples vs the exemplars (is there any difference between them?)

2- What is the point of theoretical analysis?

---

> ### Author Response · Authors · 2025-12-03
>
> We thank the reviewers for helpful comments. The choice of herding examples was motivated by their lower memory and privacy cost compared to full exemplar sets, and the theoretical analysis was intended to provide preliminary guarantees for the proposed framework. More detailed experiments and clearer explanations will be included in our future submission.

---

### Note · Authors · 2025-12-03

**Comment:**

Dear Reviewers,

We appreciate the time and effort you have dedicated to evaluating our work.

After carefully considering your comments, we have decided to withdraw our submission from ICLR 2026. We believe that addressing the issues raised will significantly improve the quality and impact of our research. We plan to undertake a more in-depth revision of the manuscript, enhancing our experimental setup with additional datasets and advanced attack methods, expanding the related works section, and correcting all identified errors and formatting issues.

Once we have thoroughly revised the paper, we intend to resubmit it for consideration in a future conference or journal. Your constructive feedback has been instrumental in guiding our improvements, and we are committed to enhancing the clarity and robustness of our work based on your suggestions.

Thank you again for your support and insightful comments.

Sincerely,

Authors

**Withdrawal Confirmation:**

I have read and agree with the venue's withdrawal policy on behalf of myself and my co-authors.